# F-VLM: Open-Vocabulary Object Detection upon Frozen Vision and Language Models

**Weicheng Kuo[*], Yin Cui[†], Xiuye Gu[†], AJ Piergiovanni[*], Anelia Angelova[*]**
[*]Google Research, Brain Team; [†]Google Research, Perception
{weicheng, yincui, xiuyegu, ajpiergi, anelia}@google.com

## Abstract

We present **F-VLM**, a simple open-vocabulary object detection method built upon **F**rozen **V**ision and **L**anguage **M**odels. F-VLM simplifies the current multi-stage training pipeline by eliminating the need for knowledge distillation or detection-tailored pretraining. Surprisingly, we observe that a *frozen* VLM: 1) retains the locality-sensitive features necessary for detection, and 2) is a strong region classifier. We finetune only the detector head and combine the detector and VLM outputs for each region at inference time. F-VLM shows compelling scaling behavior and achieves +6.5 mask AP improvement over the previous state-of-the-art on LVIS open-vocabulary detection benchmark at system level. In addition, we demonstrate very competitive results on COCO open-vocabulary detection benchmark and cross-dataset transfer detection, in addition to significant training speedup and compute savings. The code will be released [1].

## 1 Introduction

Detection is a fundamental vision task that aims to localize and recognize objects in an image. However, the data collection process of manually annotating bounding boxes or instance masks is tedious and costly, which limits the modern detection vocabulary size to an order of $10^3$. This is orders of magnitude smaller than the vocabulary humans use to describe the visual world. To overcome such limitation, we focus on open-vocabulary object detection (Zareian et al., 2021; Gu et al., 2022) to take detection beyond a fixed set of vocabulary.

Recently, vision and language models (VLMs) have gained strong open-vocabulary visual recognition capability by learning from Internet-scale image-text pairs (Radford et al., 2021; Jia et al., 2021). They are typically applied to zero-shot classification (*e.g.*, on ImageNet) using frozen weights without finetuning, which stands in stark contrast to the existing paradigms of retraining or finetuning when applying VLMs for open-vocabulary detection.

Intuitively, in order to align the image content with the text description during training, VLMs may learn locality sensitive and discriminative features that are transferable to object detection. Observations in Figure 1 support our intuition. Surprisingly, features of a frozen VLM contain rich information that are both locality sensitive for describing object shapes (col. 2) and discriminative for region classification (col. 3). This motivates us to explore using frozen VLM features for open-vocabulary detection, which entails accurate localization and classification of objects in the wild.

We propose F-VLM – a simple and scalable open-vocabulary detection approach built upon frozen VLMs. For localization, we simply attach a detector head to predict object regions. For open-vocabulary recognition, we apply the VLM feature pooler (*e.g.*, a self-attention layer) on the region features from frozen backbones at test time. We train only the detector head upon a frozen VLM backbone, and combine the detection scores with the corresponding VLM predictions at test time. Our recipe reduces the training complexity of an open-vocabulary detector to below that of a standard detector, obviating the need for knowledge distillation, detection-tailored pretraining, or weakly supervised learning. By preserving the knowledge of pretrained VLMs completely, F-VLM maintains a similar philosophy as ViTDet (Li et al., 2022c) to decouple the detector-specific learning from the more task-agnostic vision knowledge in the backbone.

---

[1]Project page: https://sites.google.com/view/f-vlm/home

| Input Image | K-Means Clustering of Frozen Features | GT Regions Classified by Frozen Features | F-VLM (Ours) |
|---|---|---|---|

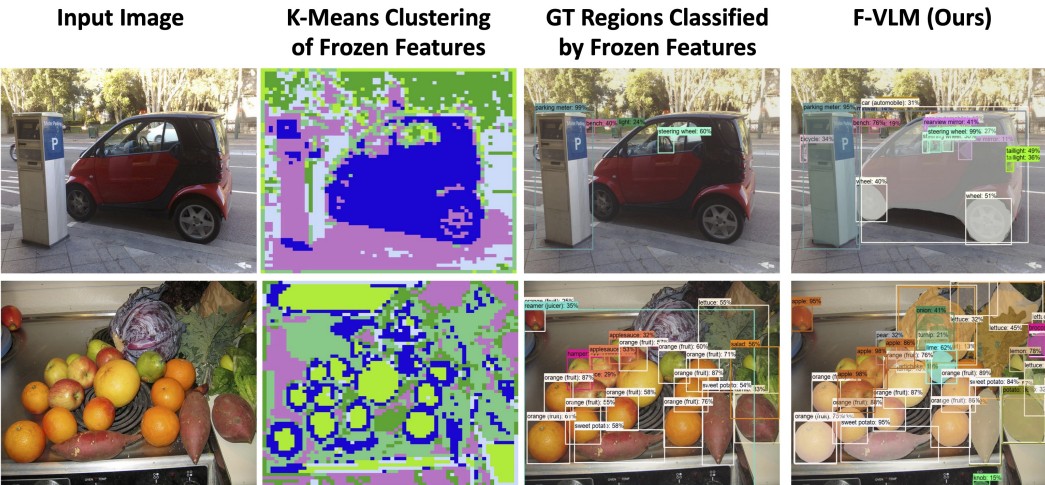

Figure 1: We explore the potential of frozen VLM (*e.g.*, CLIP) features for open-vocabulary detection. The feature grouping reveals rich semantic and locality-sensitive information where object boundaries are nicely delineated (col. 2, see Appendix C for more details). The same frozen features can classify groundtruth regions well without finetuning (col. 3). Therefore, we propose to build a open-vocabulary detector on top of a frozen VLM (col. 4) without a need for knowledge distillation, detection-tailored pretraining, or weakly supervised learning. F-VLM significantly reduces training complexity and compute requirement, and achieves the state-of-the-art performance at system level.

We demonstrate the efficacy of F-VLM on LVIS (Gupta et al., 2019), COCO (Lin et al., 2014) and Objects365 (Shao et al., 2019). Here is a summary of our contributions and observations:

- We propose F-VLM – a simple open-vocabulary detection method upon frozen VLMs without knowledge distillation, detection-tailored pretraining, or weakly supervised learning.

- Despite its simplicity, F-VLM achieves strong performance, surpassing the previous state-of-the-art on LVIS open-vocabulary detection benchmark by 6.5 mask $AP_r$ at system level and outperforming existing approaches in cross-dataset transfer (COCO, Objects365).

- F-VLM shows compelling scaling behavior with consistent performance improvements by increasing the backbone capacity (*e.g.*, +14.2 LVIS mask $AP_r$ with our largest backbone).

- F-VLM has much fewer trainable parameters, allowing it to train significantly faster. Compared with a strong open-vocabulary detection method ViLD (Gu et al., 2022), F-VLM not only achieves better performance, but also provides up to 200× training compute savings.

We hope these findings will facilitate the research community to further explore frozen VLMs for a broader range of computer vision tasks.

## 2    RELATED WORK

**Zero-shot/Open-vocabulary visual recognition and representation learning.** Zero-shot and open-vocabulary recognition has been a long-standing problem in computer vision. Earlier works use the visual attributes to represent categories as binary codebooks and learn to predict the attributes for novel categories (Jayaraman & Grauman, 2014; Rohrbach et al., 2011). DeViSE (Frome et al., 2013) and ConSE (Norouzi et al., 2014) pioneer to learn a joint image-text embedding space using deep learning. Many works have shown the promise of representation learning from natural language associated with images, such as image tags (Chen & Gupta, 2015; Divvala et al., 2014; Joulin et al., 2016) or text descriptions (Desai & Johnson, 2021; He & Peng, 2017; Sariyildiz et al., 2020; Wang et al., 2009; Zhong et al., 2021). Recently, popular large VLMs scale up by training on billions of image-text pairs and acquire strong image-text representation by contrastive learning (Radford et al., 2021; Jia et al., 2021; Pham et al., 2021; Zhai et al., 2022). These models achieve strong zero-shot performance on many classification benchmarks and show clear benefits in scaling model capacity.

While all the above works study image-level recognition, the focus of this paper is on the object-level understanding. Recently, Vasconcelos et al. (2022) has shown frozen classification models

are beneficial for closed-vocabulary detection with adequate detector head capacity. In addition, a frozen VLM can serve as a teacher model and combine with self-training for zero-shot semantic segmentation (Zhou et al., 2022a). In contrast, we study how to use frozen VLM directly as part of an open-vocabulary object detector.

**Zero-Shot/Open-vocabulary object detection.** It is costly and labor-intensive to scale up data collection and annotation for large vocabulary detection. Zero-shot detection aims to alleviate the challenge by learning to detect novel categories not present in the training data. Many methods address this by aligning the image region features to category word embeddings (Bansal et al., 2018; Rahman et al., 2020; Demirel et al., 2018; Zheng et al., 2020), or synthesizing visual features with a generative model (Hayat et al., 2020; Zhu et al., 2020). Recently, Zareian *et al.* proposes the open-vocabulary detection (OVD) benchmark with a view to bridge the performance gap between ZSD and supervised learning (Zareian et al., 2021). The model was first pretrained on image-caption data to recognize novel objects, and then finetuned for zero-shot detection (Zareian et al., 2021).

Following the OVD benchmark, ViLD (Gu et al., 2022) proposes to distill the rich representation of pretrained VLM into the detector, and DetPro (Du et al., 2022) improves upon ViLD by applying the idea of prompt optimization. RegionCLIP (Zhong et al., 2022) develops a region-text pretraining strategy that leverages pretrained VLMs and image-caption data, while Detic (Zhou et al., 2022c) jointly trains a detector with weak supervision. VL-PLM (Zhao et al., 2022) explores pseudo-labeling on unlabeled data with object proposals and VLMs for OVD. GLIP (Li et al., 2022b) formulates object detection as a phrase grounding task and pretrains on a wide variety of detection, grounding, and caption datasets for zero/few-shot object detection. Similarly, OWL-ViT (Minderer et al., 2022) proposes to finetune pretrained vision transformers on a suite of detection/grounding datasets. All mentioned methods require training the entire detector from scratch, finetuning after detection-tailored pretraining, or training on a suite of detection/grounding datasets. In contrast, F-VLM trains only the standard detector head upon a frozen VLM without using any of the above.

## 3 METHOD

### 3.1 OVERVIEW

In this paper we address the problem of open-vocabulary object detection. At training time, the model has access to the detection labels of $C_B$ base categories, but needs to detect objects from a set of $C_N$ novel categories at test time. To make the settings more practical (Zareian et al., 2021), we follow previous works and assume the availability of a pretrained vision and language model (VLM) which has learned from plenty of image-text pairs on the internet (Gu et al., 2022).

Figure 2 shows the overall F-VLM architecture. We propose to build the open-vocabulary object detector upon frozen VLMs by training only the detector head upon frozen features, which guarantees to completely preserve the open-vocabulary classification ability of pretrained VLMs. At test time, we combine the detector scores with the VLM scores to obtain open-vocabulary object detection scores. By directly using frozen pretrained models, our approach is simple and easily scalable.

### 3.2 PRETRAINING FROM VISION AND LANGUAGE MODELS

Recently, Vision and Language Models (VLM) are popular because of their rich knowledge and strong representation for both visual and linguistic domains. We desire to retain their knowledge as much as possible, in order to minimize the effort/cost to adapt the VLMs for open-vocabulary detection. Following existing works (Du et al., 2022; Gu et al., 2022; Zhong et al., 2022), we focus on contrastively pretrained VLMs in this paper *e.g.* (Jia et al., 2021; Radford et al., 2021). Contrastive VLMs typically have the image and text encoders trained jointly with a contrastive objective. We use the frozen image encoder as the detector backbone, and the text encoder for caching the text embeddings of detection dataset vocabulary offline (see Sec. 3.3).

We consider the VLM image encoder in two parts: 1) the feature extractor $\mathcal{F}(\cdot)$, *e.g.* ResNet-50 (Radford et al., 2021), and 2) the last feature pooling layer $\mathcal{P}(\cdot)$, *e.g.* an attention pooling layer (Radford et al., 2021). We adopt the same backbone architecture as the image feature extractor $\mathcal{F}(\cdot)$, which allows us to directly use the frozen weights and inherit the rich semantic knowledge (see Fig. 2a). Along with the backbone initialization, we also adopt the same image pre-processing scheme as the VLM pretraining to maintain its open-vocabulary recognition ability. We use the last VLM pooling layer $\mathcal{P}(\cdot)$ for open-vocabulary recognition at test time only (see Sec. 3.4). Build-

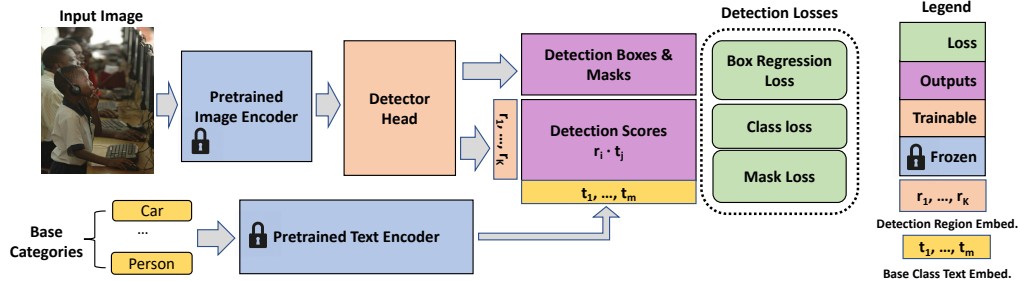

(a) **F-VLM training architecture**. At training time, F-VLM is simply a detector with the last classification layer replaced by base-category text embeddings. The detector head is the only trainable part of the system, which includes RPN (Ren et al., 2015), FPN (Lin et al., 2017), and Mask R-CNN heads (He et al., 2017).

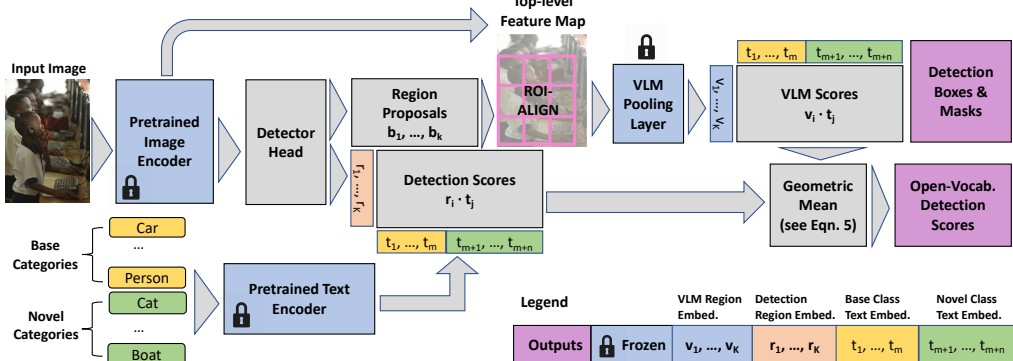

(b) **F-VLM inference architecture**. At test time, F-VLM uses the region proposals to crop out the top-level features of VLM backbone and compute the VLM score per region. The trained detector head provides the detection boxes and masks, while the classification scores are a combination of detection and VLM scores.

Figure 2: **F-VLM architecture.** We present both training and inference time architectures of F-VLM, where the VLM pooling layer and detection score combination are the differences.

ing upon the frozen backbone features, we adopt Mask R-CNN (He et al., 2017) head and feature pyramid network (Lin et al., 2017) as the detector head following previous works (Du et al., 2022; Gu et al., 2022; Zhong et al., 2022). The detector head is randomly initialized and is *the only trainable component* of F-VLM. Despite the image-level pretraining, we found empirically that the frozen VLM backbone contains adequate locality sensitive features to enable accurate downstream detection (see Appendix C).

## 3.3 TEXT-EMBEDDING REGION CLASSIFIER

**Notations:** Let's define $I$ as the input image, $\mathcal{F}(I)$ the backbone features from the image encoder. Let $\mathcal{Q}(\cdot)$ be the function that yields a region embedding $\mathbf{r}_b$ from $\mathcal{F}(I)$ and a given box region proposal $b$, which involves FPN (Lin et al., 2017), ROI-Align (He et al., 2017), and Faster R-CNN head (Ren et al., 2015). We have:

$$\mathbf{r}_b = \mathcal{Q}(\mathcal{F}(I), b) \tag{1}$$

Standard detectors use K-way classifier because the training and test time categories are the same. This design does not support the open-vocabulary settings which require new categories to be added at test time. To accommodate this, we replace the last fully connected layer with the text embeddings of base categories (see Fig. 2a). At inference time, we can simply expand the text embeddings to include novel categories for open-vocabulary detection (see Fig. 2b). An advantage of such design is that the system can generalize to the novel categories near $C_B$ in the embedding space.

To generate the text embeddings, it is critical to use the matching text encoder of the image encoder because they were pre-trained together. Apart from $C_B$, the background category is represented by a generic phrase "background" for compatibility with other categories. At training time, the proposals not matched to any groundtruth boxes in $C_B$ are treated as background. For each region, we compute the cosine similarity of $\mathbf{r}_b$ with the text embeddings of $C_B$ and "background", and apply a learnable

temperature $\tau$ on the logits. The detection scores $\mathbf{z}(\mathbf{r}_b)$ are given by:

$$\mathbf{z}(\mathbf{r}_b) = Softmax(\frac{1}{\tau}\left[cos(\mathbf{r}_b, \mathbf{t}_{bg}),\ cos(\mathbf{r}_b, \mathbf{t}_1),\ \cdots,\ cos(\mathbf{r}_b, \mathbf{t}_{|C_B|})\right]) \quad (2)$$

where $cos(\mathbf{a}, \mathbf{b}) = \mathbf{a}^\top\mathbf{b}/(\|\mathbf{a}\|\|\mathbf{b}\|)$, and $\mathbf{t}_i$ denotes the text embeddings of class $i$. We apply the standard softmax cross entropy loss on the logits (see Fig. 2a). At test time, we keep the "background" category and expand the text embeddings from $C_B$ to $C_B \cup C_N$ for open-vocabulary detection. Similar designs have been used by previous works (Zareian et al., 2021; Gu et al., 2022).

## 3.4 OPEN-VOCABULARY RECOGNITION

The ability to perform open-vocabulary recognition at region level is integral to F-VLM. Since the backbone features are frozen, they do not overfit to the base categories and can be directly cropped for region-level classification. F-VLM performs this open-vocabulary classification only at test time.

To obtain the features for a region $b$, we apply the VLM pooling layer $\mathcal{P}(\cdot)$ on the cropped backbone output features $\mathcal{F}(I)$ (see Sec. 3.2 for notations). Because the pooling layer requires fixed-size inputs, e.g. 7x7 for R50 (Radford et al., 2021), we crop and resize the region features with ROI-Align $\mathcal{R}(\cdot)$ (He et al., 2017) (see Fig. 2b). Unlike existing works (Gu et al., 2022; Du et al., 2022), we do not crop and resize the RGB image regions and cache their embeddings in a separate offline process, but train the detector head in one stage. This is simpler and more space-efficient. In addition, we do not crop VLM region features with $\mathcal{R}(\cdot)$ during training because the backbone features are frozen. Using the notations from equation 1, we obtain the VLM region embedding $\mathbf{v}_b$ by:

$$\mathbf{v}_b = \mathcal{P}(\mathcal{R}(\mathcal{F}(I), b)) \quad (3)$$

where $b$ denotes the box region and $\mathbf{v}_b$ corresponds to $v_1, ..., v_k$ in Fig. 2b. Note $\mathcal{R}(\cdot)$ is used at test time only. Similar to equation 2, we compute the VLM scores by cosine similarity as follows:

$$\mathbf{w}(\mathbf{v}_b) = Softmax(\frac{1}{T}\left[cos(\mathbf{v}_b, \mathbf{t}_{bg}),\ cos(\mathbf{v}_b, \mathbf{t}_1),\ \cdots,\ cos(\mathbf{v}_b, \mathbf{t}_{|C_{B\cup N}|})\right]) \quad (4)$$

where $T$ is a fixed temperature and the text embeddings include both the $C_B$ and $C_N$ at inference time (see Fig. 2b). We use a fixed temperature to adjust the scale of VLM scores relative to the detection scores in equation 2. In the special case when the region $b$ is equal to the whole image, the VLM scores $\mathbf{w}(\mathbf{v}_b)$ becomes equivalent to the zero-shot image classification scores.

Despite never being trained on regions, the cropped region features of $\mathcal{F}(\cdot)$ maintain good open-vocabulary recognition ability. However, we observe the cropped region features are not sensitive enough to the localization quality of the regions, i.e. a loosely vs tightly localized box both have similar features. This may be good for classification, but is problematic for detection because we need the detection scores to reflect localization quality as well. To remedy this, we apply the geometric mean to combine the VLM scores $w(\mathbf{v}_b)_i$ in equation 4 with the detection scores $z(\mathbf{r}_b)_i$ in equation 2 for each region $b$ and category $i$. The final detection scores $s(\mathbf{r}_b)_i$ are given by:

$$s(\mathbf{r}_b)_i = \begin{cases} z(\mathbf{r}_b)_i^{(1-\alpha)} \cdot w(\mathbf{v}_b)_i^{\alpha} & \text{if } i \in C_B \\ z(\mathbf{r}_b)_i^{(1-\beta)} \cdot w(\mathbf{v}_b)_i^{\beta} & \text{if } i \in C_N \end{cases} \quad (5)$$

where $\alpha, \beta \in [0, 1]$ control the VLM score weights for base/novel categories, and the background score comes directly from the detector i.e., $s(\mathbf{r}_b)_0 = z(\mathbf{r}_b)_0$. Compared to the ensemble system in (Gu et al., 2022), our design is simpler without a need for knowledge distillation or double Faster R-CNN heads. We show ablations of different score fusion designs in Appendix A.2.

## 3.5 OPEN-VOCABULARY LOCALIZATION

How to localize and separate the novel objects from the background is an important problem in open-vocabulary detection. Standard detectors are not designed for localizing novel objects because most of them apply class-specific localization, including the box regression and mask prediction heads, e.g., Mask R-CNN (He et al., 2017). Inspired by the learned objectness (Kim et al., 2022; Kuo et al., 2015; Wang et al., 2020), we use *class-agnostic* box regression and mask prediction heads instead. That is, for each region proposal, we predict one box and one mask for all categories, rather than one

per category. This simple change allows us to localize novel objects in the open-vocabulary settings. We note that F-VLM framework is not specific to the choice of Mask R-CNN detector head and other models can potentially be applied as well *e.g.* (Carion et al., 2020; Redmon et al., 2016). We choose Mask R-CNN per existing works (Gu et al., 2022; Zareian et al., 2021; Zhong et al., 2022).

## 4 EXPERIMENTS

**Implementation Details.** We choose Mask R-CNN (He et al., 2017) with feature pyramid network (Lin et al., 2017) as our detector head throughout the paper. The head design follows (Ghiasi et al., 2021; Gu et al., 2022). We train the model for 46.1k iterations with 1024x1024 image size, large scale jittering (Ghiasi et al., 2021), batch size 256, weight decay 1e-4, momentum 0.9, and an initial learning rate 0.36. For the score combination, we use $\alpha = 0.35$ and $\beta = 0.65$ in equation 5. We use a maximum of 300 detections per image, and set temperature $T = 0.01$ in equation 4. We use CLIP (Radford et al., 2021) prompt templates and take the average text embeddings of each category. Please refer to Appendix G for a full list of hyper-parameter configurations.

### 4.1 OPEN-VOCABULARY DETECTION BENCHMARK

**LVIS Benchmark.** We evaluate our approach on the LVIS dataset (Gupta et al., 2019) which contains a large and diverse set of 1203 object categories suitable for open-vocabulary detection. Following the existing works (Gu et al., 2022; Zhong et al., 2022), we treat the frequent and common categories as the base categories $C_B$ for training, and hold out the rare categories as novel categories $C_N$ for testing. Mask $AP_r$ is the main metric we benchmark on. To ensure reproducibility, we report the mean of 5 independent runs following the protocol of (Gu et al., 2022) and the best practice of LVIS challenge (Gupta et al., 2019). For fair comparison, we adopt the same Mask R-CNN head architecture as (Gu et al., 2022) and use the same large scale jittering recipe (Ghiasi et al., 2021).

Table 1 presents our results on LVIS. In the R50 comparison, F-VLM ranks second among the other alternatives based on knowledge distillation, pretraining, or joint training with weak supervision. The leading DetPro (Du et al., 2022) shows the effectiveness of prompt optimization (Zhou et al., 2022b) which can potentially benefit F-VLM too. In the system-level comparison, we observe the performance of F-VLM scales up nicely with frozen model capacity, even though the amount of trainable parameters is fixed. Our best model achieves 32.8 $AP_r$, which is +14.2 $AP_r$ from the R50 baseline and the best published results on this benchmark to our knowledge. Compared to the best existing approach (ViLD-EN-B7), we outperform by 6.5 mask $AP_r$ on the novel categories (and +5.6 overall mask AP). We provide additional results using standard 1x/3x training recipes (Wu et al., 2019) in Appendix B, where F-VLM shows similarly strong performance on shorter recipes and smaller batch size by using frozen backbones.

**COCO Benchmark.** Many existing works on zero-shot detection (Bansal et al., 2018) and open-vocabulary detection (Zareian et al., 2021; Gu et al., 2022; Zhong et al., 2022) benchmark on COCO. This setup divides COCO vocabulary into 48 base categories for training and 17 novel categories for testing. We follow the standard practice and report results in the generalized detection settings without instance segmentation. The main metric is AP50 of novel categories. Similar to LVIS, we report the mean of 5 independent runs to ensure reproducibility.

Due to the smaller number of training categories, we observe a tendency to overfit when we re-use the same LVIS training recipe. F-VLM does not rely on additional objectives *e.g.* knowedlge distillation or weak supervision to counter-balance overfitting. We therefore reduce the training epoch, background weight, and increase the weight decay to mitigate this. Please refer to Appendix G for a full list of hyper-parameters.

Table 2 shows that F-VLM is very competitive among the published results. Compared to the leading RegionCLIP (Zhong et al., 2022) which uses additional caption pretraining, F-VLM directly uses a frozen CLIP backbone. In fact, F-VLM significantly surpasses the CLIP-R50 pretrained version of RegionCLIP, which does not leverage pretraining on caption data. Compared to other approaches, F-VLM offers better performance without the use of detection-tailored pretraining, weakly supervised learning, or knowledge distillation.

**Training Resource Benchmark.** We explore the benefits of frozen VLMs in terms of training resource savings. We benchmark with ViLD (Gu et al., 2022) as it is most comparable to F-VLM. Both adopt the same Mask R-CNN head configuration and training recipe (Ghiasi et al., 2021),

Table 1: **LVIS Open-Vocabulary Object Detection Benchmark**. F-VLM outperforms the best existing approach by 6.5 mask AP on novel categories. All methods use the same instance-level supervision from LVIS (Gupta et al., 2019) base categories, CLIP (Radford et al., 2021) pretraining, and fixed prompt templates unless noted otherwise. [†]: Pretraining with CC-3M (Sharma et al., 2018). [‡]: Prompt optimization (Zhou et al., 2022b) and SoCo pretraining (Wei et al., 2021). [*]: Joint training with IN-21k (Deng et al., 2009). [⋆]: ALIGN model (Jia et al., 2021).

| Backbone (# Params) | Pretrained CLIP | Method | Distill | Trainable Backbone | $AP_r$ | AP |
|---|---|---|---|---|---|---|
| R50 Comparison: | | | | | | |
| R50 | ViT-B/32 | ViLD (Gu et al., 2022) | ✓ | ✓ | 16.1 | 22.5 |
| R50 | ViT-B/32 | ViLD-Ens. (Gu et al., 2022) | ✓ | ✓ | 16.6 | 25.5 |
| R50 | ViT-B/32 | DetPro (Du et al., 2022)[‡] | ✓ | ✓ | 19.8 | 25.9 |
| R50 | ViT-B/32 | Detic-ViLD (Zhou et al., 2022c)[*] | ✗ | ✓ | 17.8 | 26.8 |
| R50 | R50 | RegionCLIP (Zhong et al., 2022)[†] | ✓ | ✓ | 17.1 | 28.2 |
| R50 | R50 | F-VLM (Ours) | ✗ | ✗ | 18.6 | 24.2 |
| System-level Comparison: | | | | | | |
| R152 (60M) | ViT-B/32 | ViLD (Gu et al., 2022) | ✓ | ✓ | 18.7 | 23.6 |
| R152 (60M) | ViT-B/32 | ViLD-Ens. (Gu et al., 2022) | ✓ | ✓ | 18.7 | 26.0 |
| EN-B7 (67M) | ViT-L/14 | ViLD-Ens. (Gu et al., 2022) | ✓ | ✓ | 21.7 | 29.6 |
| EN-B7 (67M) | EN-B7[⋆] | ViLD-Ens. (Gu et al., 2022) | ✓ | ✓ | 26.3 | 29.3 |
| R50 (26M) | ViT-B/32 | DetPro-Cascade (Du et al., 2022)[‡] | ✓ | ✓ | 20.0 | 27.0 |
| R50 (26M) | ViT-B/32 | Detic-CN2 (Zhou et al., 2022c)[*] | ✗ | ✓ | 24.6 | 32.4 |
| R50x4 (87M) | R50x4 | RegionCLIP (Zhong et al., 2022)[†] | ✓ | ✓ | 22.0 | 32.3 |
| ViT-L/14 (303M) | ViT-L/14 | OWL-ViT (Minderer et al., 2022) | ✗ | ✓ | 25.6 | 34.7 |
| R50x4 (87M) | R50x4 | F-VLM (Ours) | ✗ | ✗ | 26.3 | 28.5 |
| R50x16 (167M) | R50x16 | F-VLM (Ours) | ✗ | ✗ | 30.4 | 32.1 |
| R50x64 (420M) | R50x64 | F-VLM (Ours) | ✗ | ✗ | **32.8** | 34.9 |

Table 2: **COCO Open-Vocabulary Object Detection Benchmark.** F-VLM is very competitive with the other methods trained with various sources. All methods use the ResNet50 backbone (He et al., 2016; Radford et al., 2021). RegionCLIP additionally use COCO Captions[†] (Lin et al., 2014) or CC3M[‡] (Sharma et al., 2018) for pretraining. [⋆]: CLIP initialization without region-level pretraining. [*]: Joint training with COCO captions.

| Method | Training source | Novel AP | AP |
|---|---|---|---|
| WSDDN (Bilen & Vedaldi, 2016) | image-level labels in $C_B \cup C_N$ | 19.7 | 19.6 |
| Cap2Det (Ye et al., 2019) | | 20.3 | 20.1 |
| ZSD (Bansal et al., 2018) | instance-level labels in $C_B$ | 0.31 | 24.9 |
| DELO (Zhu et al., 2020) | | 3.41 | 13.0 |
| PL (Rahman et al., 2020) | | 4.12 | 27.9 |
| OVR-CNN (Zareian et al., 2021) | image captions in $C_B \cup C_N$ instance-level labels in $C_B$ | 22.8 | 39.9 |
| CLIP-RPN (Gu et al., 2022) | CLIP image-text pairs instance-level labels in $C_B$ | 26.3 | 27.8 |
| ViLD (Gu et al., 2022) | | 27.6 | 51.3 |
| Detic[*] (Zhou et al., 2022c) | | 27.8 | 45.0 |
| RegionCLIP[‡] (Zhong et al., 2022) | | **31.4** | 50.4 |
| RegionCLIP[†] (Zhong et al., 2022) | | 26.8 | 47.5 |
| RegionCLIP[⋆] (Zhong et al., 2022) | | 14.2 | 42.7 |
| F-VLM (Ours) | | 28.0 | 39.6 |

and neither require detection-tailored pretraining. We follow ViLD and compare the training cost on TPUv3 cores on the same batch size. The data about ViLD training time and resource use is obtained directly from the authors (Gu et al., 2022). To keep the benchmark simple, we assume the pretrained VLMs are given and exclude their training costs from the comparison. For F-VLM, we use the R50x64 backbone and report the average over 5 independent runs.

Table 3: **Training Resource Benchmark.** We report LVIS mask $AP_r$ to show the performance vs training cost trade-off. F-VLM can outperform ViLD (Gu et al., 2022) with 226× less compute.

| Method | Mask $AP_r$ | #Iters | Epochs | Training Cost (Per-Core-Hour) | Training Cost Savings |
|---|---|---|---|---|---|
| ViLD-EN-B7 (Gu et al., 2022) | 26.3 | 180k | 460 | 8000 | 1× |
| F-VLM (Ours) | 32.8 | 46.1k | 118 | 565 | 14× |
| F-VLM (Ours) | 31.0 | 5.76k | 14.7 | 71 | 113× |
| F-VLM (Ours) | 27.7 | **2.88k** | **7.4** | **35** | **226×** |

Table 3 shows that F-VLM can achieve top performance with much less compute. Compared to the state-of-the-art (ViLD-EN-B7) at system level, F-VLM can achieve better performance with only 7.4 epochs of training, which is 226× more compute-efficient and 57× faster in wall clock time. We believe the efficiency gain arises from the frozen backbone, which substantially simplifies the learning process. This is orthogonal to the detection-tailored pretraining used by existing works to speed up training (Zareian et al., 2021; Zhong et al., 2022). Apart from resource savings, F-VLM has potential for substantial memory savings at training time by running the backbone in inference mode (see Appendix E for more details). F-VLM system runs almost as fast as a standard detector (He et al., 2017) at inference time, because the only addition is a single attention pooling layer (Radford et al., 2021) on the detected region features (see Fig. 2b).

**Transfer Detection Benchmark.** We explore the potential of F-VLM as a general-purpose detector for different data sources with a view to move towards non dataset-specific detection. F-VLM trained on one dataset can be directly applied to another by swapping out the vocabulary without any finetuning, *e.g.*, replacing the 1203 LVIS categories with COCO 80 categories. The models we use are trained on LVIS base categories and tested on COCO and Objects365-v1 validation splits following the transfer setup of ViLD (Gu et al., 2022). Since COCO and Objects365 have smaller vocabularies than LVIS, category and image overlaps are hard to avoid. We calculate the vocabulary overlap between COCO/Objects365 and LVIS base categories to be 91% and 63% respectively. Please refer to Appendix D for more discussion about this benchmark.

Table 4 presents the results in comparison with prior works and supervised baselines. We observe the performance of F-VLM improves steadily as we scale up the frozen model capacity. On Objects365/COCO, the best F-VLM outperforms existing works ViLD by +3.2/+5.9 and DetPro by +4.9/+5.6, closing the gap with a supervised model on COCO (-33%) and Objects365 (-40%). The results are reported in Box AP averaged over 5 runs. There is no longer distinction between base and novel categories in the transfer setting, so we assume all categories are novel and use $\beta$ alone to combine detection and VLM scores in equation 5 (see Sec. 3.4). We observe that only the detection scores are needed ($\beta = 0$) for COCO, while the optimal $\beta = 0.3$ to $0.4$ on Objects365.

Table 4: **Transfer detection of F-VLM.** We evaluate LVIS-trained F-VLM on COCO and Objects365 without finetuning. F-VLM demonstrates strong scaling property with a gain of +7.3/+5.8 AP on COCO/Objects365 by increasing backbone capacity.

| Method | COCO | | | Objects365 | | |
|---|---|---|---|---|---|---|
| | AP | $AP_{50}$ | $AP_{75}$ | AP | $AP_{50}$ | $AP_{75}$ |
| Supervised (Gu et al., 2022) | 46.5 | 67.6 | 50.9 | 25.6 | 38.6 | 28.0 |
| ViLD-R50 (Gu et al., 2022) | 36.6 | 55.6 | 39.8 | 11.8 | 18.2 | 12.6 |
| DetPro-R50 (Du et al., 2022) | 34.9 | 53.8 | 37.4 | 12.1 | 18.8 | 12.9 |
| F-VLM-R50 (Ours) | 32.5 | 53.1 | 34.6 | 11.9 | 19.2 | 12.6 |
| F-VLM-R50x4 (Ours) | 36.0 | 57.5 | 38.7 | 14.2 | 22.6 | 15.2 |
| F-VLM-R50x16 (Ours) | 37.9 | 59.6 | 41.2 | 16.2 | 25.3 | 17.5 |
| F-VLM-R50x64 (Ours) | **39.8** | **61.6** | **43.8** | **17.7** | **27.4** | **19.1** |

## 4.2 ANALYSES AND VISUALIZATION

**Ablations.** We present ablation studies on backbone finetuning, score fusion design/parameters, feature pyramid capacity, and background weight in Appendix A. Here we summarize our findings.

In the exploration of finetuning vs frozen backbone (see Table 5), we discover that finetuning improves the standard detection (base categories) but slightly hurts the open-vocabulary detection

(novel categories). It remains an open question whether more sophisticated finetuning strategies can benefit open-vocabulary detection.

In the score fusion studies (see Table 6), we observe that geometric mean is significantly better than the arithmetic mean (+8 $AP_r$). It is likely because the geometric mean requires a high-scoring region to have good detection and VLM scores simultaneously, whereas the arithmetic mean may favor regions with high detection or VLM scores. In Table 7a and Table 7b, we study the score fusion weights and observe that $\beta = 0.65$ and $\alpha = 0.35$ are most beneficial (see equation 5). Neither detection nor VLM scores alone are sufficient as $\beta = 0, 1$ both yield sub-optimal performances. In Table 7c, we study the temperature in equation 4 and find the optimal $T = 10^{-2}$ is much smaller than the value of learnable $\tau \approx 1.0$ at the end of training (see equation 2). This highlights the need to use a separate $T$ for VLM scores instead of using $\tau$ for both detection and VLM scores.

In Table 8, we explore the effects of increasing feature pyramid capacity to enhance the representation learned upon the frozen backbone features. Our results show that larger pyramid benefits standard detection (base categories) without compromising the open-vocabulary detection (novel categories). In Table 9, we study the influence of background weights by Zareian et al. (2021); Zhong et al. (2022) on open-vocabulary detection and found it to help slightly (0.1 to 0.5 $AP_r$). Please refer to Appendix A for the full experimental results.

**Detection Visualization.** Figure 3 visualizes F-VLM on open-vocabulary detection and transfer detection tasks. The transfer detection is done by replacing the dataset vocabulary without finetuning. On LVIS and Objects365 (Shao et al., 2019), F-VLM correctly detects both novel and common objects. Please see Appendix H for more details and visualization.

A key benefit of open-vocabulary detection is to test on out-of-distribution data with categories given by users on the fly. Thus, we apply F-VLM to Ego4D (Grauman et al., 2022), a real-world ego-centric application. Despite the large domain shift, F-VLM is able to detect many novel and common objects. Please see Appendix I for more details and visualization.

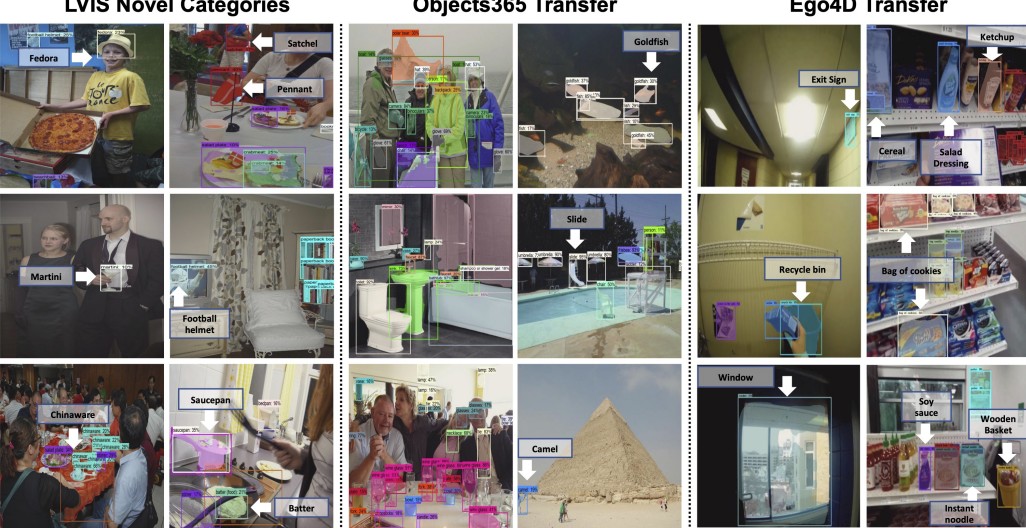

Figure 3: **F-VLM open-vocabulary and transfer detections**. 1-2nd col.: Open-vocabulary detection on LVIS. We only show the novel categories for clarity. 2-4th col.: Transfer detection on Objects365. 4-6th col.: Transfer detection on Ego4D. Novel categories detected: *fedora, martini, pennant, football helmet* (LVIS); *camel, slide, goldfish* (Objects365); *exit sign, recycle bin, window, soy sauce, wooden basket, cereal, bag of cookies, instant noodle, salad dressing, ketchup* (Ego4D).

## 5 CONCLUSION

We present F-VLM – a simple open-vocabulary detection method built upon *frozen* VLMs without a need for knowledge distillation, detection-tailored pretraining, or weakly supervised learning. F-VLM offers significant training speedup and compute savings, achieves the new state-of-the-art on LVIS benchmark at system level, and shows very competitive transfer detection. We hope this study can help the community explore frozen VLMs for a wider range of vision tasks.

## 6 REPRODUCIBILITY STATEMENT

We plan to open source the code for reproducibility. We have provided the model, experimental and implementation details in the paper or the supplemental materials (Section 4 and Section G). The CLIP model (Radford et al., 2021), Mask R-CNN model (He et al., 2017), and all datasets used (Lin et al., 2014; Gupta et al., 2019; Shao et al., 2019; Grauman et al., 2022) in this work are publicly available.

## 7 ETHICS STATEMENT

We demonstrate new capabilities in detecting previously unseen categories of objects, and particularly on challenging benchmarks and transfer settings. Our models utilize the rich information embedded in Vision-Language Models, which may reinforce deficiencies and biases in the internet data and propagate potentially harmful biases or stereotypes. The models we trained are used here for evaluation/benchmark purposes and need more rigorous probing for bias, fairness, *etc.*, before using them for any other purpose.

## 8 ACKNOWLEDGMENTS

We thank our colleagues at Google Research for their advice and helpful discussion.

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

APPENDIX

## A ABLATION

### A.1 FINETUNING VERSUS FROZEN BACKBONE

We explore the pros and cons of backbone finetuning compared to the frozen backbone. We observe that finetuning the backbone with the same training recipe diverges, so we apply gradient clipping (max norm = 1.0) and reduce the backbone learning rate significantly. Table 5 shows that although finetuning can benefit the base categories, it slightly compromises the novel category with higher memory/compute footprint.

Table 5: **Finetuning vs frozen backbone.** Finetuning does not benefit the novel categories ($AP_r$) but improves the base categories ($AP_c$, $AP_f$).

| Backbone LR | $AP_r$ | $AP_c$ | $AP_f$ | AP |
|---|---|---|---|---|
| 1e-3 | 18.1 | 25.7 | 30.2 | 26.2 |
| 1e-4 | 18.1 | 24.9 | 28.8 | 25.3 |
| 0.0 | 18.6 (+0.5) | 24.0 | 26.9 | 24.2 |

### A.2 SCORE FUSION

We explore the use of arithmetic vs geometric means to fuse the VLM and detection scores in equation 5. Table 6 shows that using geometric mean is significantly better by more than 8 points. In Figure 4, we perform a dense grid sweep over $\alpha, \beta$ and confirm the 8-point gap between geometric and arithmetic means still holds.

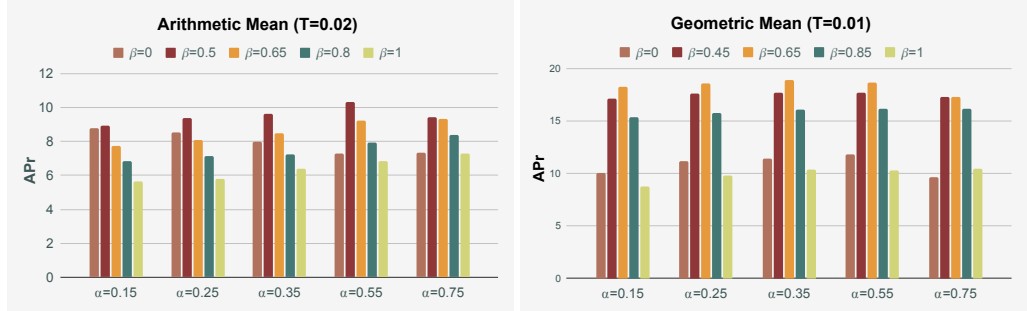

Figure 4: **Hyper-parameter sweep on score fusion parameters**. We observe that geometric means (right) are significantly better than arithmetic means (left). All results are based on a trained F-VLM R50 model.

We perform a more in-depth study of score fusion parameters in Table 7. From the table, we see that $\beta$ is the main tunable parameter of our model, and the performance is relatively robust to $\alpha$. For most practical use cases, we recommend setting $T = 0.01$. The temperature $\tau$ in Equation 2 is learned automatically and needs no tuning.

### A.3 FEATURE PYRAMID CAPACITY

We explore the effects of increasing the feature pyramid (Lin et al., 2017) capacity to enhance the representation learned upon the frozen backbone features. To increase the FPN capacity, we simply repeat the lateral and top-down connections of FPN $N$ times before applying the post-hoc convolution. We insert a ReLU and BatchNorm layer in each lateral connection, and add a skip connection from the backbone feature maps to every level (*i.e.* $1, 2, ..., N$). The post-hoc convolution and BatchNorm layers are kept the same. All runs are repeated 5 times and trained for 46.1k steps following the same protocol as the LVIS benchmark of the manuscript.

Table 6: **Score Fusion.** We study different score fusion mechanisms of F-VLM. We report $AP_r$ and AP on LVIS. Geometric mean is significantly more effective than arithmetic mean. All results are average over 5 independent runs using R50 backbone.

| Fusion | $\beta$ | $\alpha$ | $T$ | $AP_r$ | AP |
|--------|------|------|------|-----------|------|
| Arithmetic | 0.65 | 0.35 | 0.01 | 9.1 | 16.4 |
| Arithmetic | 0.65 | 0.35 | 0.02 | 9.3 | 19.8 |
| Arithmetic | 0.5 | 0.75 | 0.02 | 10.3 | 15.9 |
| Geometric | 0.65 | 0.35 | 0.01 | 18.6 (+8.3) | 24.2 |

Table 7: **Score Fusion Parameters**. We study different geometric mean fusion parameters by comparing their $AP_r$ on LVIS. All results are based on a trained F-VLM R50 model. Default settings are in gray .

(a) **VLM-score weights for novel classes.** We fix $\alpha = 0.35$ and $T = 0.01$.

| $\beta$ | $AP_r$ | AP |
|------|--------|------|
| 0.0 | 11.3 | 22.9 |
| 0.45 | 17.8 | 24.2 |
| 0.65 | **18.9** | **24.2** |
| 0.85 | 16.1 | 22.5 |
| 1.00 | 10.3 | 17.8 |

(b) **VLM-score weights for base classes.** We fix $\beta = 0.65$ and $T = 0.01$.

| $\alpha$ | $AP_r$ | AP |
|------|--------|------|
| 0.15 | 18.2 | 24.3 |
| 0.25 | 18.8 | 24.5 |
| 0.35 | **18.9** | **24.2** |
| 0.55 | 18.7 | 22.4 |
| 0.75 | 17.3 | 17.8 |

(c) **Temperature for VLM logits.** We fix $\beta = 0.65$ and $\alpha = 0.35$

| $T$ | $AP_r$ | AP |
|------|--------|------|
| 0.0025 | 14.9 | 18.7 |
| 0.005 | 17.3 | 22.2 |
| 0.01 | **18.9** | **24.2** |
| 0.02 | 17.8 | 24.8 |
| 0.04 | 13.9 | 24.4 |

Table 8 shows that increased feature pyramid capacity improves the base categories significantly ($AP_c$, $AP_f$) without compromising the novel categories ($AP_r$). Although it is common to improve the base categories at the cost of novel categories, enlarged feature pyramid leads to improvements on all categories including a slight improvement of 0.1 on $AP_r$.

Table 8: **Feature Pyramid Capacity.** We observe that larger feature pyramid improves the base categories ($AP_c$, $AP_f$) without compromising the novel categories ($AP_r$).

| Backbone | Feature Pyramid | $AP_r$ | $AP_c$ | $AP_f$ | AP |
|----------|-----------------|--------|--------|--------|------|
| R50x64 | FPN (Lin et al., 2017) | 32.8 | 35.4 | 35.4 | 34.9 |
| R50x64 | FPN ($N = 12$) | 32.9 (+0.1) | 37.5 (+1.9) | 38.3 (+2.9) | 37.0 (+2.1) |

### A.4 BACKGROUND WEIGHT

We study the effects of background weight (Zareian et al., 2021; Zhong et al., 2022) on F-VLM. Consistent with the findings in (Zhong et al., 2022), we found that a background weight of 0.9 is slightly better than the default 1.0 in Table 9. Therefore, we use background weight 0.9 as default. All results are average over 3 independent runs.

## B COMPUTATION-FRIENDLY TRAINING

To facilitate comparison with the broader research community, we validate the efficacy of F-VLM in more computation-friendly $1\times$ (12 epochs) and $3\times$ (36 epochs) settings (Wu et al., 2019) by using smaller batch size and no large-scale-jittering (LSJ) augmentation (Ghiasi et al., 2021). The results are listed in Table 10.

We observe that F-VLM is robust to the number of training epochs, batch size, with/without LSJ (Ghiasi et al., 2021) for both the smallest and largest backbones. This stands in contrast to the sensitivity of fully supervised learning to these hyper-parameters, and is consistent with our findings in Table 3 that frozen backbone contributes to the training efficiency and stability.

Table 9: **Background Weight.** We study the effects of background weight of F-VLM on LVIS.

| Backbone | Background Weight | $AP_r$ |
|----------|-------------------|--------|
| R50 | 1.0 | 18.3 |
| R50 | 0.9 | 18.4 (+0.1) |
| R50x64 | 1.0 | 32.4 |
| R50x64 | 0.9 | 32.9 (+0.5) |

Table 10: **Benchmark on computation-friendly training recipes.** By leveraging frozen backbone, F-VLM is robust to shorter schedule and smaller batch size. All results are reported as the average over 5 runs. Our default settings are in gray .

| Backbone | LSJ | # Epochs | Batch Size | $AP_r$ |
|----------|-----|----------|------------|--------|
| R50 | | 12 (1×) | 16 | 18.1 |
| R50 | | 36 (3×) | 64 | 18.5 |
| R50 | ✓ | 100 | 256 | 18.6 |
| R50x64 | | 12 (1×) | 16 | 31.9 |
| R50x64 | | 36 (3×) | 64 | 32.6 |
| R50x64 | ✓ | 100 | 256 | 32.8 |

## C  EXPLORING THE STRUCTURE OF FROZEN FEATURES

To understand the effectiveness of F-VLM, we perform k-means clustering to probe the structures present in the frozen VLM features (*e.g.* CLIP). We use CLIP R50x4 backbone and LVIS dataset for visualization. Only the last layer output features are used for clustering, because these features can be used for zero-shot region classification at the same time. Figure 5 demonstrates that the features form nice clusters around salient objects of the scenes (*e.g.*, skis, motorbikes, people), and naturally separate object parts (*e.g.*, donut toppings, bus wheels) without explicit supervision. We believe these emergent properties of frozen VLM are promising avenues to push open-vocabulary detection beyond the domains of existing detection datasets.

## D  ANALYSIS OF TRANSFER DETECTION BENCHMARK

Many existing works benchmark transfer detection across common detection datasets (Gu et al., 2022; Du et al., 2022; Li et al., 2022b; Minderer et al., 2022). In particular, the LVIS to COCO and Objects365 transfer detection is a recent benchmark proposed by Gu et al. (2022). We analyze and report the vocabulary overlap percentage between COCO/Objects365 and LVIS base categories in Table 11.

Table 11: **Transfer Detection Vocabulary Overlap.** We observe substantial overlap between COCO and Objects365 vocabulary and LVIS base categories.

| Method | COCO | Objects365 |
|--------|------|------------|
| Name matching | 85% | 56% |
| Name matching + near duplicate removal | 91% | 63% |

Simple name matching shows clear overlap, while the removal of near duplicates (e.g. synonyms and non-alphabetic character removal) reveals even more. In addition, we notice that COCO has more overlap than Objects365 due to its smaller vocabulary. These results show the limitation of existing transfer detection setup, and we encourage the community to move towards larger transfer detection benchmarks with less vocabulary overlap.

| Image | K=3 | K=6 | Image | K=3 | K=6 |

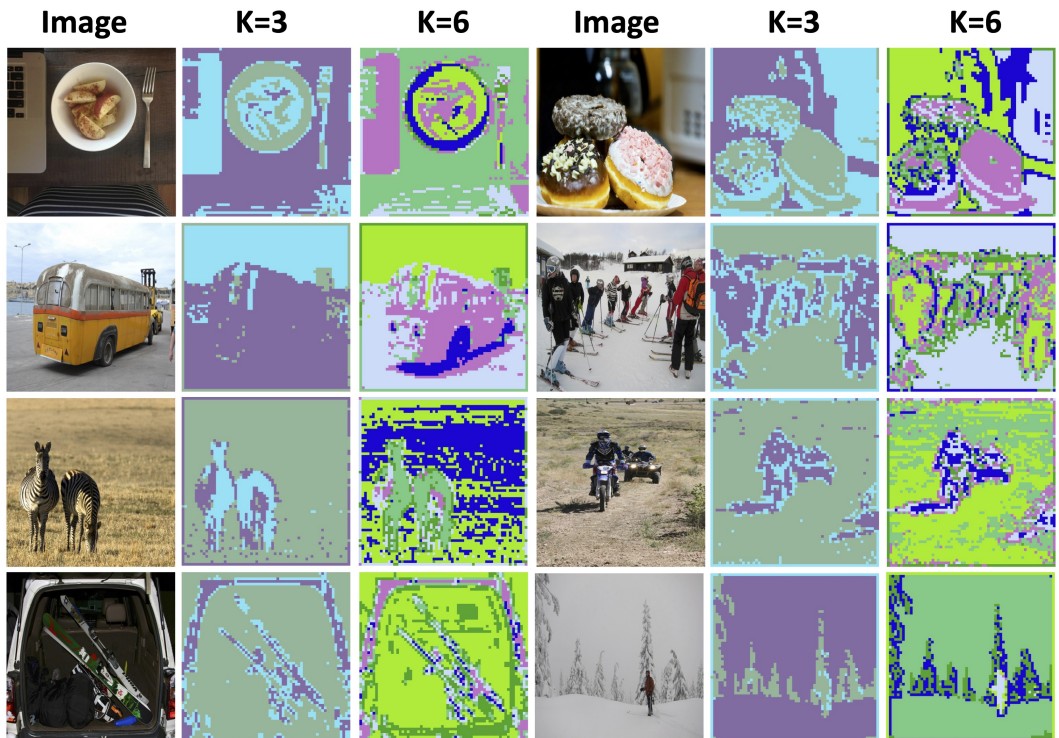

Figure 5: **Understanding the frozen VLM feature clusters**. Salient objects and object parts emerge naturally from the clustering of frozen VLM features.

## E   MEMORY USE

The memory consumption of F-VLM is almost the same as Mask R-CNN, with all class-specific heads changed to class-agnostic. F-VLM uses the same batch size as Gu et al. (2022) in the default settings, although we found it works well even with much smaller batch size and epoch length (see Appendix B).

Moreover, F-VLM has significant memory saving potential compared to existing approaches that fine-tune the backbone, especially with large backbones. At training time, F-VLM does not need to store forward-pass activations, gradients or gradient moments, and the memory use of the backbone is just the backbone weights plus a small amount of current activations. This makes F-VLM highly memory efficient especially with large backbones. In practice, the actual memory use depends on the low-level implementation of each deep learning library. In pytorch, for example, "torch.no_grad()" context manager [2] can enable such behavior.

## F   CAN WE USE OTHER VLMS?

In this work, we adopt the widely used CLIP (Radford et al., 2021) to develop a simple open-vocabulary detection recipe based on frozen backbones. Moving forward, we believe it would be very interesting to explore different pretrained VLMs, which may involve substantial modifications to F-VLM. For example, ViT-based pretrained VLMs (Radford et al., 2021; Li et al., 2021; 2022a; Zhai et al., 2022) have become popular recently. These VLMs require single-scale ViT-based detector such as ViTDet (Li et al., 2022c) to adapt them for open-vocabulary detection. To use VLMs like ALBEF (Li et al., 2021) and BLIP (Li et al., 2022a), it is important to consider how to efficiently compute all-pair region-text similarities with multimodal encoders (as opposed to dual encoders). Furthermore, it is an open question how to use captioning VLMs (Wang et al., 2021; Hu et al., 2022)

---

[2]https://pytorch.org/docs/stable/generated/torch.no_grad.html

or masked multimodal VLMs (Singh et al., 2022) for open-vocabulary detection. We believe these are all interesting directions for the community to explore.

## G  IMPLEMENTATION DETAILS

Table 12 summarizes the hyper-parameters we use for LVIS and COCO experiments. On LVIS, we adopt the same hyper-parameters as Gu et al. (2022) except for a shorter schedule (due to frozen backbone) and a background weight following Zareian et al. (2021); Zhong et al. (2022) (see A.4). The hyper-parameter differences on COCO are to mitigate overfitting to the ZSD-COCO split (Bansal et al., 2018) of 48 categories, which is significantly smaller than the 800 LVIS base categories. This is necessary because F-VLM does not use other objectives *e.g.* knowledge distillation or weak supervision to counter-balance overfitting.

Table 12: **F-VLM hyper-parameter configuration.**

| Configuration | LVIS | COCO |
|---|---|---|
| Optimizer | SGD | SGD |
| Momentum | $\beta = 0.9$ | $\beta = 0.9$ |
| Weight decay | 1e-4 | 1e-2 |
| Gradient Clipping | none | none |
| Learning rate (LR) | 0.36 | 0.02 |
| Step decay factor | $0.1\times$ | $0.1\times$ |
| Step decay schedule | [0.8, 0.9, 0.95] | [0.9, 0.95, 0.975] |
| Warmup LR / steps | 3.2e-3 / 1k | 3.2e-3 / 1k |
| Total Steps | 46.1k | 11.25k |
| Batch size | 256 | 64 |
| Epochs | 118 | 6 |
| Augmentation | LSJ (Ghiasi et al., 2021) | LSJ (Ghiasi et al., 2021) |
| NMS Threshold | 0.5 | 0.4 |
| Base VLM weight $\alpha$ | 0.35 | 0.2 |
| Novel VLM weight $\beta$ | 0.65 | 0.45 |
| Background weight $\gamma$ | 0.9 | 0.2 |
| VLM Temperature $T$ | 0.01 | 0.01 |

We observe some differences in the optimal hyper-parameters for different backbone architectures. With the R50x64 backbone, we notice an improvement of 1.0 $AP_r$ when we use $T = 0.02$ as opposed to the default $T = 0.01$. For R50 backbone, we notice an improvement of 0.5 $AP_r$ when we apply a gradient clipping of 1.0 maximum gradient norm as opposed to none. We report the performance using the optimal settings for these two backbones.

## H  VISUALIZATION

We visualize more F-VLM outputs on LVIS novel categories and transfer detection to Objects365 in Figure 6. On LVIS, F-VLM is able to correctly detect many rare categories including baguet, neckerchief, tabasco sauce, and gourd. On Objects365, F-VLM can detect many categories in complex scenes including ducks, traffic sign, street light, and air conditioner. These confirm that our approach is effective for novel category detection and transfer detection to another dataset. We use the R50x4 backbone for this visualization. The model was trained on the LVIS base categories following the main benchmark of the paper.

## I  APPLICATION ON EGO-CENTRIC DATA

A key benefit of open-vocabulary detection is to test on out-of-distribution data with categories given by users on the fly. Thus, we apply F-VLM to Ego4D (Grauman et al., 2022), a real-world ego-centric application. We train F-VLM on a mixture of full LVIS, Objects365, and COCO datasets to expand its training vocabulary for application in the wild, and use the R50x16 backbone for this

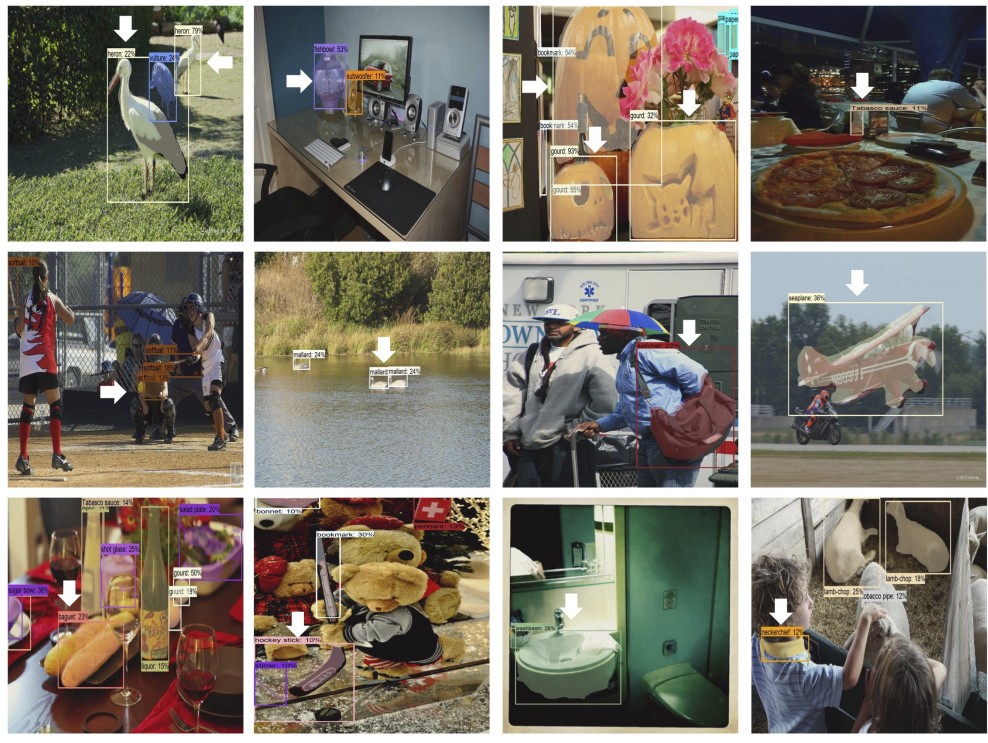

(a) **LVIS novel category detection**. F-VLM can detect many novel categories despite its simplicity using a frozen VLM. The white arrows point to the novel objects correctly detected by F-VLM. For clarity, we only show the novel categories.

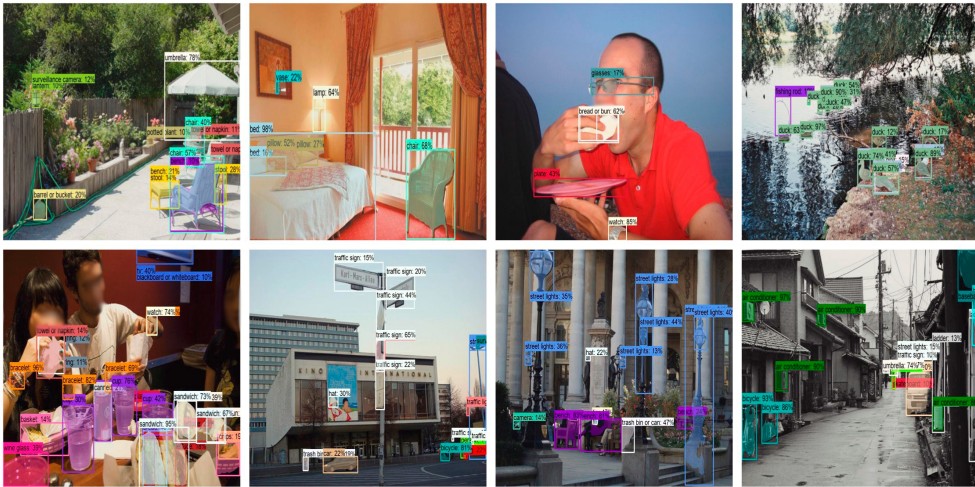

(b) **Objects365 transfer object detection**. F-VLM can be applied to a new dataset and detect many challenging categories without further finetuning.

Figure 6: F-VLM Visualization on LVIS novel categories and Objects365 transfer object detection.

experiment. The model is not trained on Ego4D in order to evaluate for transfer detection. The categories are provided by the user based on visual inspection of the video.

For the indoor scene, the category names provided by the user are as follows: *plate, cabinet, stove, towel, cleaning rag, ventilator, knob, sauce and seasoning, steel lid, window, window blinds, plant, light switch, light, door, carpet, exit sign, doormat, hair, door lock, tree, poster on the wall, sticker on the wall, faucet, recycle bin, rack, hand, can, carton, trash, Christmas tree, plastic container, fridge*.

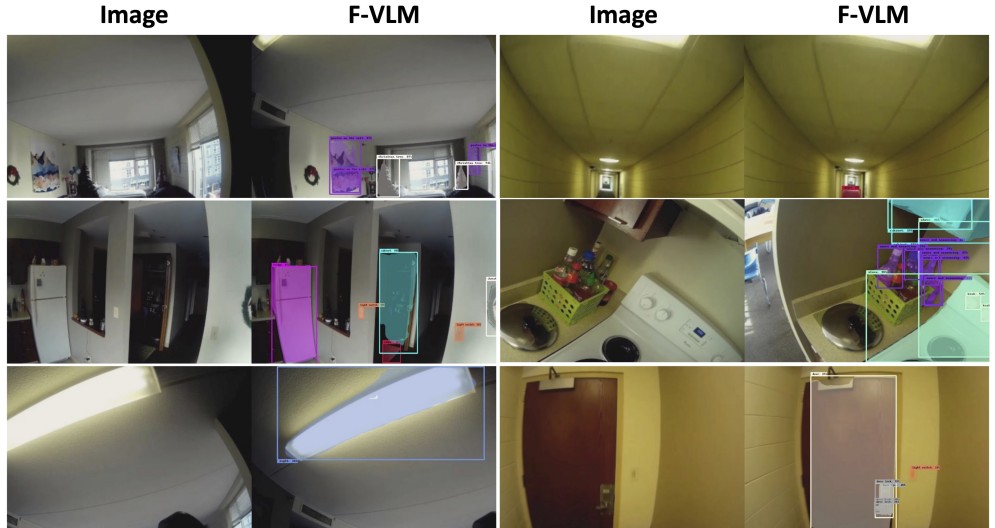

(a) **Indoor scene.** Under the challenging viewing angles, occlusion, and lighting conditions, F-VLM still manages to detect many objects in the scene.

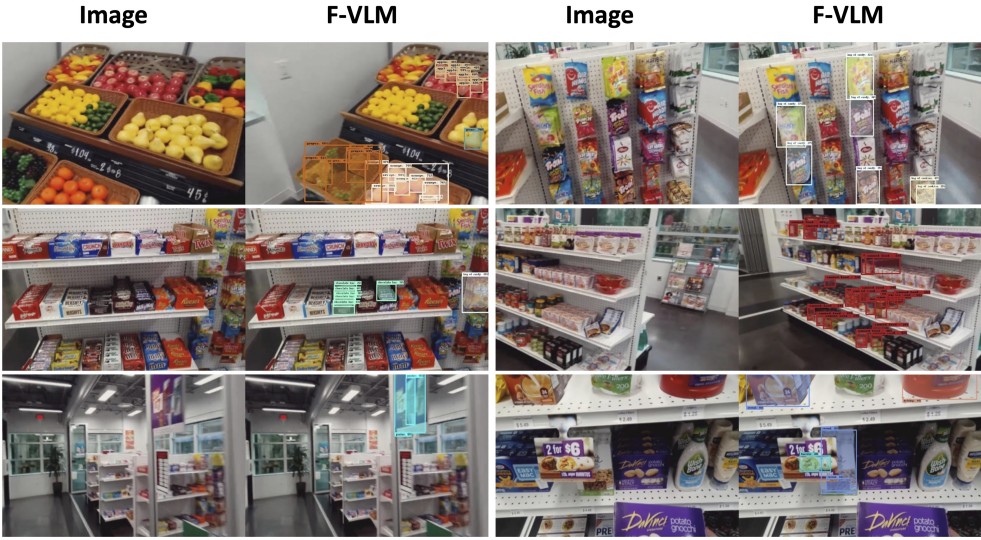

(b) **Grocery store scene**. The scene is very crowded with a wide variety of objects. F-VLM is able to detect many of them.

Figure 7: F-VLM Visualization on Ego4D (Grauman et al., 2022) transfer object detection. Novel categories detected: *light switch, light, door lock, sauce and seasoning, bag of candies, canned food, and burrito*.

For the grocery store scene, the category names provided by the user are as follows: *exit sign, poster, chocolate bar, bag of candy, bag of cookies, snack, oreo, soy sauce, apple, pear, orange, grapes, price tag, cereal, instant noodle/ramen, cracker, ATM machine, instant noodle, wooden basket, red ramen bowls, magazine, drugs and medicine, Mayo, Ketchup, Cup noodle, burrito, Lays/Sun chips, seasoning sauce, black carton, salad dressing, canned food*.

Figure 7 shows that F-VLM is able to detect many objects in the ego-centric videos despite the large domain shift and challenging viewing conditions. In particular, it is able to detect many novel categories not present in the training set, such as light switch, light, door lock, sauce and seasoning, bag of candies, canned food, and burrito.

