# OpenReview forum: "Open-Vocabulary Object Detection upon Frozen Vision and Language Models"
_ICLR.cc/2023/Conference — ICLR 2023 poster_

### Official Review · Reviewer_e3Lv · 2022-10-18

**Confidence:** 5
**Correctness:** 3
**Technical Novelty And Significance:** 3
**Empirical Novelty And Significance:** 4
**Recommendation:** 8

**Clarity, Quality, Novelty And Reproducibility:**

- The paper is well written and easy to follow. The figures complement the text well.
- The main contribution can be summarized as leveraging VLMs for OVD in a more effective way than prior works (like knowledge distillation in ViLD or finetuning in RegionCLIP or VL-PLM). I think this is a valid contribution, simple but effective, as stated above.
- Regarding reproducibility, the authors plan to release code, but also the list of hyper-parameters in the appendix is good.
- The last paragraph of page 3 mentions that the same image pre-processing is used as in the VLM, CLIP in this case. I assume that the image resizing/cropping to 224x224 is not included. And I assume the input image size is also the reason only the CNN-based backbones of CLIP are used in this work, but not the ViT-based backbones. Is that assumption correct?
- The text of the detection labels in Figure 3 is too small. Please increase the font size in the detection visualization.

**Strength And Weaknesses:**

Strengths:
- The proposed approach is simple, but highly effective considering the reduced training compute requirements. The paper shows a novel way how to leverage pre-trained VLMs for OVD. I think the reduced compute requirements are interesting to the research community, particularly for institutions with limited compute budgets.
- The comparison with prior works is good; using standard benchmarks in a fair setting.
- I think the ablation studies are good; I like the summary of findings where details and tables are put into the appendix.

Weaknesses:
- Given that the VLM is frozen, I think the paper would benefit from evaluating different choices of the VLM, beyond the for ResNet-variants in CLIP. For instance, ALBEF [A] or BLIP [B] are publicly available.
- Table 4 and the corresponding analysis about the training compute requirements is great, but I would find it much better to make this analysis more prominent in the paper (instead of the last item after ablation studies). After all, this is one of the key arguments this paper tries to convey: keeping the VLM frozen, you only need to train detector-specific parameters and still get good results with significantly reduced training compute. While I expect future methods to achieve better results by training the whole model (maybe better finetuning and directly including image-caption data), the training efficiency of this work would still stand.
- A recent, but concurrent paper from ECCV [C] could be included in the discussion, because it also uses a "frozen" VLM, but for creating pseudo labels (the "fast" version also applies ROI-align on CLIP-image-encoder features).
- I understand that the cross-dataset experiment (LVIS to COCO and Objects-365) has been used in prior work, but I do not think this is a great experimental setup to claim any generalization. As mentioned in the paper, the overlap of categories is high ... also evident from the chosen hyper-parameters (beta = 0). First, I would suggest to explicitly state the amount of overlap of the label spaces to make this point more clear. Second, I think it is better to move this experiment into the appendix (the reader can then still see the comparison with prior work), and instead propose novel ways to demonstrate generalization. I like the evaluation on Ego4D. Is a quantitative evaluation possible? An alternative could be the ODinW benchmark [D]. Although the localization ability is not evaluated, one could try to use ImageNet for such label space generalization experiments, with the assumption that the highest scoring bounding box per image is chosen as the prediction.
- What about the memory consumption for the proposed model? Table 3 does not mention this aspect. Related to that, was the batch size increased compared to the ViLD training settings?

References:
* [A] Align before Fuse: Vision and Language Representation Learning with Momentum Distillation. Li et al. NeurIPS 2021
* [B] BLIP: Bootstrapping Language-Image Pre-training for Unified Vision-Language Understanding and Generation. Li et al. ICML 2022
* [C] Exploiting Unlabeled Data with Vision and Language Models for Object Detection. Zhao et al. ECCV 2022
* [D] ODinW benchmark https://eval.ai/web/challenges/challenge-page/1839/overview

**Summary Of The Paper:**

The paper is about open-vocabulary object detection (OVD), where detection models are trained from a set of base categories with bounding box annotations, as well as a set of image-caption pairs. Like prior works, this paper leverages existing, pre-trained vision & language models (VLM) like CLIP that were trained on large quantities of image-caption pairs. Unlike prior works, this work shows that an object detector can be put on top of the frozen VLM. Specifically, the proposed object detector consists of the frozen VLM as the backbone, while detector-specific layers (region proposals, feature pyramid network, and region classification) are trainable. The final classification of each region is a combination (geometric mean) of detector scores and off-the-shelf VLM scores. The paper demonstrates that state-of-the-art results can be achieved with a frozen VLM, while the training compute requirements can be reduced significantly.

**Summary Of The Review:**

Overall, I think the paper should be accepted. Although the technical contribution is simple, the proposed framework is effective and gives state-of-the-art results. The paper is well written and the authors plan to release code.

---

> ### Author Response · Authors · 2022-11-18
> **Response to Reviewer e3Lv. Thank you for the review. (Part 2 / 2)**
>
> >**Image size and preprocessing**
>
> Yes, the image crop/resize operation to 224x224 is not used in F-VLM, because we use 1024x1024 input image size for detection. The other preprocessing operations e.g. image normalization are still used.
> You’re correct that the ability to generalize across very different image sizes (e.g. from 224 to 1024) is important for F-VLM. We focus on CNN backbones as they provide a simple setting to explore the ideas of frozen VLMs. On the other hand, ViT-based backbones require more studies on how to adapt positional embeddings and the single-scale backbone for detection. We feel that how to use frozen ViT-based VLMs could be an interesting direction for future explorations.
>
> >**Figure 3 detection label text**
>
> Thank you for the feedback. We have updated Figure 3 to include large-font text labels to indicate the detected novel objects.

---

> > ### Comment · Reviewer_e3Lv · 2022-11-25
> > **Response to author feedback**
> >
> > Thank you for the detailed answers, additional explanations, and the revisions on the paper. Very much appreciated.

---

> ### Author Response · Authors · 2022-11-18
> **Response to Reviewer e3Lv. Thank you for the review. (Part 1 / 2)**
>
> Thank you for the helpful review. We carefully address your questions and comments below and have updated the submission pdf accordingly.
>
> >**Evaluate different VL models**
>
> Thank you for the valuable suggestion. Our motivation for this work is to leverage strong pretrained VLMs. We adopt the widely used CLIP backbones, and set our first milestone to be achieving the state-of-the-art at system level. Our next goal is to explore different VLMs which involve non-trivial modifications to our current recipes. For example, we are currently exploring ViT-based pretrained VLMs [CLIP, ALBEF, BLIP] which require a new detector like ViTDet. To use ALBEF and BLIP for open-vocabulary detection, we need to address additional challenges of efficiently computing all-pair region-text similarities with multimodal encoders (as opposed to the commonly used dual encoders). We have included a discussion of exploring different VLMs for open-vocabulary detection in Appendix F.
>
> >**Highlight training efficiency**
>
> We appreciate your recognition of our computation savings by leveraging frozen models directly. We have moved the old Table 4 (now Table 3) and the training resource benchmark section to right after the main open-vocabulary detection benchmark to highlight this more. Thanks for the suggestion.
>
> >**Related work on pseudo-labeling**
>
> Thanks for sharing this concurrent work. We agree the idea of applying RoIAlign on CLIP image features to create pseudo labels is quite relevant. We have included a description of [C] in the Related Works.
>
> >**Cross-dataset label space overlap**
>
> Thanks for the valuable suggestion. We follow existing works to benchmark on transfer detection among common detection datasets [ViLD, DetPro, GLIP, OwL-ViT]. The LVIS to COCO and Objects365 transfer detection is a very recent benchmark initially used by ViLD and followed by others. We calculate the vocabulary overlap percentage between COCO/Objects365 and LVIS base categories below:
>
> |Method | COCO | Objects365 |
> |---------|:---------:|:---------:|
> | Name match | 85\% | 56\% |
> | Name match + near duplicate removal | 91\% | 63\% |
>
> Simple name matching shows clear overlap, while the removal of near duplicates (e.g. synonyms and non-alphabetic character removal) reveals even more. In addition, we notice that COCO has more overlap than Objects365 due to its smaller vocabulary. These results show the limitation of this cross-dataset transfer setup. We agree it is important to come up with alternative out-of-domain datasets for evaluation.
>
> Per reviewer’s suggestion, we have stated the amount of label space overlap in the transfer detection section to make this limitation clearer. In addition, we have included the above table and a discussion about this in Appendix D with the view to help the community move towards better transfer detection benchmarks with less vocabulary overlap.
>
> >**Cross-dataset quantitative evaluation**
>
> Thanks for finding our evaluation on Ego4D meaningful. Unfortunately, the Ego4D dataset does not provide object detection benchmarks or annotations for a proper transfer detection evaluation. Thank you for the suggestion about ODinW and ImageNet as alternatives. We agree it is very valuable to demonstrate generalization with more quantitative evaluation, and will try our best to include it in the final version.
>
> >**Memory consumption**
>
> The memory consumption of F-VLM is almost the same as Mask R-CNN, with all class-specific heads changed to class-agnostic. F-VLM uses the same batch size as ViLD in the default settings, but we found that it works well even with much smaller batch size and epoch length (see Appendix B).
>
> We note that F-VLM has significant memory saving potential at training time compared to existing approaches that fine-tune the backbone, especially with large backbones. At training time, F-VLM does not need to store forward-pass activations, gradients or gradient moments, and the memory use of the backbone is just the backbone weights plus a small amount of current activations. This makes F-VLM highly memory efficient.
>
> In practice, the actual memory use depends on the low-level implementation of each deep learning library. In PyTorch, for example, `torch.no_grad()` context manager can enable such behavior (See documentation [here](https://pytorch.org/docs/stable/generated/torch.no_grad.html)). We have included a discussion about memory use in Appendix E.

---

### Official Review · Reviewer_x64o · 2022-10-23

**Confidence:** 5
**Correctness:** 3
**Technical Novelty And Significance:** 3
**Empirical Novelty And Significance:** 2
**Recommendation:** 8

**Clarity, Quality, Novelty And Reproducibility:**

## Clarity
Good, the main contribution and ideas are clearly delivered

## Quality
The paper is well accomplished and the claims are well supported.

## Novelty
The idea can be treated as a more elegant combination of a traditional detector and a frozen vision language model for region classification. From this perspective, a natural idea is to share the vision feature extraction process, i.e., to adopt the frozen vision encoder as the backbone.

## Reproducibility
Although the implementation details are provided, the settings in not affordable and applicable to most of the university labs (large-scale image input that cost large GPU memory with a long training schedule and large batch size that requires many GPUs). Therefore, reproducibility is questioned.

**Strength And Weaknesses:**

## Strengths:
1. The paper is well-written and easy to follow
2. The proposed method is simple, fast and computing-saving for training, free-from knowledge distillation and weakly supervised learning,  while effectively exploiting the VLMs’ ability for open-vocabulary region classification.
3. The paper provides detailed ablative studies on the F-VLM

## Weakness:
1. The performance of the proposed method is inferior to existing methods under fair comparison. E.g. “R50 Comparison” in Table 1, Table 2, and “F-VLM-R50 (Ours)” in Table 3. Propmt optimization and SoCo pre-training are not orthogonal to F-VLM because it seems that these two strategies seem to be not directly applicable to F-VLM (F-VLM uses fronzon VL models), while DetPro can scale its methods to use ViT-L or R50x64. The paper also adopts heavier data augmentation and longer training schedule.
2. The insights are limited. Basically, the paper only finds a way and tells users that using a frozen VL model in the detector could achieve compatible results with scalability. But the paper could discuss more the effects of VL models, especially those of different publically available pre-trained VL models.

## Questions:
1. The training recipe (Implementation Details at Sec 4, page 6) of the detectors in the paper is not affordable and thus not widely adopted in the community (large-scale jittering, image size 1024x1024 and batch size 256) and might be unfair for comparison. What about training on a regular 12-epoch or 36-epoch schedule with batch size 16?
2. The ablation study on score fusion in Sec 4.3. What about classifying regions solely on the RoiAligned features or solely on the embeddings from the detector head?
3. In the 6th row of the 2nd paragraph of Sec 3.4, what’s the meaning of “but train the detector head with $R(\cdot)$ online in a single stage”? From what the context described, the RoiAlign $R(\cdot)$ mentioned here is only used at inference time.

**Summary Of The Paper:**

This paper presents F-VLM, which tackles open-vocabulary detection based on frozen pre-trained vision-language models (VLMs). The authors observe that frozen VLMs retain locality-sensitive features and are strong region classifiers. By only training a detection head upon frozen VLMs and utilizing regional VLM features for open-vocabulary classification at inference time, the F-VLM achieves SOTA on LVIS benchmark as well as competitive results on COCO benchmark and cross-dataset transferring. In addition, F-VLM shows compelling scalability and achieves significant training speed-up and computing saving.

**Summary Of The Review:**

Overall, the paper explores a new direction that uses frozen vision-language models for detection and provides useful experimental results which reveal the potential and scalability of this direction. The unfair comparison might be an issue and it could be better if the paper modulates its claims.

---

> ### Author Response · Authors · 2022-11-18
> **Response to Reviewer x64o. Thank you for the review. (Part 2 / 2)**
>
> >**Computation-friendly Training Benchmark**
>
> Thank you for the helpful advice. In this paper, we followed ViLD to adopt the large-scale-jittering (LSJ) recipe (Ghiasi et al 2021), which is also in Detectron2. Per reviewer’s suggestion, to facilitate comparison with the broader research community, we validate the efficacy of F-VLM in more computation-friendly 1$\times$ (12 epochs) and 3$\times$ (36 epochs) settings by using smaller batch size and no LSJ augmentation. The results show strong performance as below (the baseline 100-epoch recipe is listed for reference):
>
> |Backbone | LSJ Aug. | # of Epoch  |  Batch Size  |  AP$_r$ |
> |:---------:|:---------:|:---------:|:-----------:|:-----------:|
> |R50         |  | 12    | 16      | 18.1 |
> |R50         |  | 36    | 64      | 18.5 |
> |R50         | $\checkmark$ | 100    | 256      | 18.6 |
> |R50x64   |  | 12    | 16      | 31.9 |
> |R50x64   |  | 36    | 64      | 32.6 |
> |R50x64   | $\checkmark$ | 100    | 256      | 32.8 |
>
> All results are reported as an average over 5 runs. We observe that F-VLM is robust to the number of training epochs, batch size, with/without LSJ for both the smallest and largest backbones. This stands in contrast to the sensitivity of fully supervised learning to these hyper-parameters, and is consistent with our findings in Table 3 (Training Resource Benchmark) that frozen backbone contributes to the training efficiency and stability. We have included these results and discussion in Appendix B.
>
> >**Classify by VLM or detector features alone**
>
> Appendix A.2 Table 7 (a) shows that using either features alone achieve sub-optimal performance. Here $\beta$ controls the score fusion weight between VLM RoI features and detector head embeddings. $\beta=0$ uses the detector head embeddings only, whereas $\beta=1$ uses VLM RoI features only. The table shows that it is important to fuse the scores of both e.g. $\beta=0.65$.
>
> >**Clarification about Sec 3.4, 2nd paragraph, row 6**
>
> Thank you for the clarification. The $\mathcal{R}(\cdot)$ here refers to the ROI-ALIGN layer used in the Mask R-CNN head. At test time, we apply a separate $\mathcal{R}(\cdot)$ to extract the VLM features in addition to the $\mathcal{R}(\cdot)$ mentioned in this sentence. To avoid confusion with the $\mathcal{R}(\cdot)$ at inference time, we have removed “with $\mathcal{R}(\cdot)$ online” in this sentence as it is simply part of the detector head. We have included clarifications about this right before and after Equation 3.

---

> > ### Comment · Reviewer_x64o · 2022-12-10
> > **Response to Author Feedback**
> >
> > Thanks for the feedback. The discussions of each point are insightful and it would be great if they could be updated in the new version of the paper.

---

> ### Author Response · Authors · 2022-11-18
> **Response to Reviewer x64o. Thank you for the review. (Part 1 / 2)**
>
> Thank you for the helpful review. We carefully address your questions and comments below and have updated the submission pdf accordingly.
>
> >**Benchmark settings:**
>
> Thank you for the feedback. Per your advice, we have carefully revised all state-of-the-art statements throughout the paper to specify “at system level”.
>
> In Table 2, we presented two separate comparisons. The R50 comparison allows us to compare on the standard settings (with ViT-B/32 or R50 pretrained CLIP models), while the system-level comparison reports the top-line performance of each method. The community has used the system-level comparison for reporting state-of-the-art results, showing scalability, and ensuring the improvements on standard settings hold on larger scales. The two settings are complementary to one another and commonly used in academic benchmarks.
>
> In R50 comparison, F-VLM is highly competitive and only second to DetPro, which extends ViLD with prompt optimization. Compared to ViLD, F-VLM further simplifies the training process by using frozen models instead of knowledge distillation. We think the improvement techniques of DetPro can potentially benefit F-VLM similar to how they benefit ViLD. In addition, we have not applied federated loss which has been shown beneficial [Detic, OwL-ViT] and can likely improve our R50 baseline. Finally, F-VLM uses the R50 pretrained CLIP model (by design), which is smaller and less competitive in image classification than the ViT-B/32 CLIP model used by other methods. Given the scope of this paper, we choose to keep the system simple and comparable to the ViLD baseline, and focus on exploring scalability to larger backbones. This training simplicity allows us to focus on scaling F-VLM above the scale of existing works and achieve superior performance with affordable computation costs. That being said, we agree it would still be useful to explore these improvement techniques on top of F-VLM recipes.
>
> The study of scaling behavior is also motivated by the trend in foundation model research where model capacity is key to performance. We hope F-VLM could serve as a simple recipe to adapt those models for open-vocabulary detection and instance segmentation tasks. Please refer to the “Computation-friendly Training Benchmark” section for studies on shorter training schedules and weaker data augmentation. Thanks for bringing this up for discussion.
>
> >**Prompt optimization, SoCo pre-training, DetPro scaling**
>
> F-VLM uses a set of fixed prompt templates similar to ViLD. As DetPro improves upon ViLD by prompt optimization, we think F-VLM can potentially benefit from the same idea. On the pretraining side, F-VLM uses the CLIP image-text pretraining which is at image-level only. It’s possible that incorporating region-aware pretraining objectives into CLIP can improve the representation for F-VLM. We think these are interesting directions for future studies. Per your advice, we have removed the statement about prompt optimization and SoCo pretraining being orthogonal to avoid any confusion.
>
> We agree DetPro would likely benefit from larger backbones like ViT-L or R50x64. The paper currently reports the state-of-the-art performance on R50. More empirical results are needed to demonstrate DetPro’s scalability to larger backbones.
>
> >**Effects of VL models**
>
> Thanks for the good suggestion. In this paper we focus on contrastively pre-trained VL models because they lend themselves easily to the detection/segmentation tasks and have been adopted by many existing open-vocabulary detection/segmentation works. Exploring other kinds of pretrained VL models e.g. multimodal encoder, captioning model, masked image/language models, would be very interesting for future studies. We have added a discussion on this in Appendix F.

---

### Official Review · Reviewer_kxyx · 2022-10-23

**Confidence:** 4
**Correctness:** 3
**Technical Novelty And Significance:** 3
**Empirical Novelty And Significance:** 3
**Recommendation:** 8

**Clarity, Quality, Novelty And Reproducibility:**

The overall paper is clearly written and easy to follow.

Regarding novelty, I'll rely on my fellow reviewers' comments.

**Strength And Weaknesses:**

Strengths:
1. The overall framework is conceptually simple yet effective. Directly using a frozen pre-trained vision and language model is easier to deal with than performing knowledge distillation. And also the training cost is significantly lower than others because the entire CLIP model is frozen. It is inspiring for future work.

2. Strong quantitative results are reported on LVIS and COCO. Qualitative cross-dataset generalization results on Ego4D are also shown.

Weaknesses:
1. There are four hyper parameters to tune in the proposed approach: $\tau$ in Eq.(2), $T$ in Eq.(4), and $\alpha$ and $\beta$ in Eq.(5). They can be tuned on a validation set. But in real-world use, it is hard to carefully tune them all.

2. The paper claims that a bronze VLM "retains the locality-sensitive features necessary for detection". But only qualitative results are shown in Fig. 1 and 4. The paper implicitly assumes that the localization module (regional proposal generation) can generalize well from base categories to novel ones with quantitative verification. To validate this assumption, we can check the recall of the model on both base and novel categories.

**Summary Of The Paper:**

This paper proposes to use a frozen vision and language pre-trained model (CLIP) for open-vocabulary object detection. The image encoder from CLIP is used as the backbone and the textual encoder from CLIP is used for region proposal classification. A trainable detector head is added to generate region proposals (for localization), which are then classified into a set of base and novel categories.

**Summary Of The Review:**

Overall, this paper provides an appealing approach for open-vocabulary object detection. I'll update my rating according to other reviewers' comments and the authors' responses.

I'd like to encourage the authors to further address the following items (they are not the weaknesses, or at least not unique in this paper, so I didn't list them above).

1. According to Eq.(3), the RoIAlign operator is used in both the training and inference phases. Adding the illustration of RoIAlign in Fig. 2(a) will probably make it clearer.

2. For comparisons with different models, especially the system-level comparison, it may be a good idea to list the number of parameters of different models. So we can clearly see the relationship of the models' capacity and their accuracy.

3. The qualitative results of cross-dataset generalization is great. But the quantitative evaluation is not a "in-the-wild" setting. Basically, the region classification is dependent on the cosine similarities of a region's feature w.r.t. to a set of candidate categories. What if the vocabulary of the novel categories is very large with a lot of class names that are not available in an image? Would the accuracy of the model significantly decrease as more and more such distracting categories added? It would be great to see such comparisons (it does not require re-training the model).

4. Only certain values of $\alpha$ and $\beta$ values are studied in Table 6. Is it possible to report more ablation results and visualize them using plots?

---

> ### Author Response · Authors · 2022-11-18
> **Response to Reviewer kxyx. Thank you for the review.**
>
> Thank you for the helpful review. We carefully address your questions and comments below and have updated the submission pdf accordingly.
>
> >**Hyper-parameter tuning:**
>
> We agree that tuning multiple hyper-parameters carefully in the real-world application without a validation set can be challenging. We note that $\tau$ in Eq. (2) is a learnable parameter that users do not need to tune. Among the remaining hyper-parameters, we recommend setting T=0.01 for most practical use cases. Table 7(b) shows that the performance is relatively robust to $\alpha$, while $\beta$ is the main tunable parameter of our model. Thanks for raising this point. We have included this discussion in Appendix A.2.
>
> >**Region proposal generalization:**
>
> Thanks for the valuable feedback! We agree that the recall of region proposals on base and novel categories can inform us of the generalization ability of frozen VLM features. The results are presented below:
>
> |Backbone | AR$_\text{rare}$@100  |  AR$_\text{rare}$@300  |  AR$_\text{rare}$@1000 | AR$_\text{base}$@100  |  AR$_\text{base}$@300  |  AR$_\text{base}$@1000 |
> |---------|:---------:|:---------:|:-----------:|:-----------:|:-----------:|:-----------:|
> |R50       | 31.4    | 40.8      | 50.7     | 33.3    | 42.1      | 49.2     |
> |R50x64       | 34.5    | 44.6      | 52.4     | 37.3    | 45.7      | 52.1     |
>
> We observe that the recall on base and rare (novel) categories are very similar overall. For a smaller number of proposals i.e. 100, the recall of base categories is slightly higher because the training occurs on base categories only. The gap quickly shrinks in AR@300 and disappears in AR@1000. This observation holds for both R50 (smallest) and R50x64 (largest) backbone sizes.
>
> >**RoIAlign operator:**
>
> Thanks for the clarification. The RoIAlign operator in Eq. (3) is only used in the inference phase for VLM region embedding extraction but not in the training phase as in Fig 2(a). In the training phase, the RoIAlign operator is used only inside the Mask RCNN detector head. The RoIAlign is not needed to crop VLM region embeddings at training time because we keep the backbone features frozen. We have included a sentence to clarify this right before and after Eq. (3).
>
> >**List number of parameters:**
>
> Thanks for the good suggestion. We agree it will be informative to see the relationship between model capacity and accuracy, and have added the backbone capacity in Table 1 for the system-level comparison.
>
> >**Distracting categories:**
>
> Thank you for your comments on cross-dataset generalization. We would like to clarify that we do not just use the groundtruth categories at inference time, but the whole LVIS vocabulary of 1203 classes. Thus, the model has to reject a large number of distracting categories that are not present in the image. The reason we focus on the LVIS benchmark in this work is to maximize the number of novel categories (c.f. COCO), which reflects this scenario. We would be more than happy to explore if larger detection datasets become available in the future.
>
> The challenge of distracting categories is common to most open-vocabulary detectors. We expect most approaches to struggle when distracting categories similar to the target categories are added. We agree this would be an interesting study and will try our best to include it in the final version.
>
> >**More ablations in Table 6:**
>
> We report more ablations on a dense grid of $\alpha$ and $\beta$ values using the same fusion mechanisms in Table 6, and visualize them with bar charts in Figure 4 of Appendix A.2. We observe that the gap between arithmetic and geometric means remains large and choose geometric means for score fusion.

---

### Public Comment · ~Jincheng_Li1 · 2023-02-03
**code re-implementation**

Impressive work.
To follow it, we re-implemented the codes of the paper but could not achieve the results.
There must be some details we overlooked.
We especially look forward to the code released by the authors, and then, we can double-check our re-implemented code via code comparison.
Many thanks for the open source and have a nice day!

---

### Decision · Program_Chairs · 2023-01-20

**Decision:**

Accept: poster

**Justification For Why Not Higher Score:**

Method is too simple to merit oral.

**Justification For Why Not Lower Score:**

No reviewers recommend rejection; two confidently recommend acceptance.

**Metareview: Summary, Strengths And Weaknesses:**

This paper shows that using a pretrained VL model and only fine-tuning the detection head is very effective and efficient. The reviewers appreciate the simplicity and effectiveness of the method, shown on two main and two additional datasets. They raise concerns about limited analyses and insights, aspects of the experimental setting, and the difficulty of tuning hyperparameters, and request additional metrics and VL methods in place of CLIP. Overall the importance of the findings outweighs the concerns.

**Note From Pc:**

if the above contains the word "oral" or "spotlight" please see: "oral" presentation means -> notable-top-5% and "spotlight" means -> notable-top-25%. As stated in our emails, we are disassociating presentation type from AC recommendations